# The Microzone Structure Regulation of Diamond/Cu-B Composites for High Thermal Conductivity: Combining Experiments and First-Principles Calculations

**DOI:** 10.3390/ma16052021

**Published:** 2023-02-28

**Authors:** Zhongnan Xie, Wei Xiao, Hong Guo, Boyu Xue, Hui Yang, Ximin Zhang, Shuhui Huang, Mingmei Sun, Haofeng Xie

**Affiliations:** 1State Key Laboratory of Nonferrous Metals and Processes, GRINM Group Co., Ltd., Beijing 101400, China; 2GRIMAT Engineering Institute Co., Ltd., Beijing 101400, China; 3General Research Institute for Nonferrous Metals, Beijing 100088, China

**Keywords:** metal matrix composites, diamond/Cu composites, interface thermal conductance, microzone structure, first-principles calculations

## Abstract

The interface microzone characteristics determine the thermophysical properties of diamond/Cu composites, while the mechanisms of interface formation and heat transport still need to be revealed. Here, diamond/Cu-B composites with different boron content were prepared by vacuum pressure infiltration. Diamond/Cu-B composites up to 694 W/(mK) were obtained. The interfacial carbides formation process and the enhancement mechanisms of interfacial heat conduction in diamond/Cu-B composites were studied by HRTEM and first-principles calculations. It is demonstrated that boron can diffuse toward the interface region with an energy barrier of 0.87 eV, and these elements are energetically favorable to form the B_4_C phase. The calculation of the phonon spectrum proves that the B_4_C phonon spectrum is distributed in the range of the copper and diamond phonon spectrum. The overlapping of phonon spectra and the dentate structure together enhance the interface phononic transport efficiency, thereby improving the interface thermal conductance.

## 1. Introduction

Heat dissipation has become a bottleneck in the development of electronic packaging technology because of the increasing power density of chips. Diamond-reinforced copper matrix composites are considered to be promising for heat dissipation of high-power chips, due to their excellent thermophysical properties and tunable thermal expansion [1]. However, diamond is naturally neither wettable nor reactive with Cu, even under high temperature, which is not conducive to gain sound interfacial bonding for thermal conduction [2,3].

Interface modification is an effective method to improve the interface wettability remarkably. Introduction of carbide-forming elements in composites, such as B, Cr, Ti and W [4,5,6,7], can transform the interfacial bonding into chemical bonding, which could lead to a significant increase in thermal conductivity as well as interfacial bonding strength. By adding a tungsten coating on the diamond filler, Abyzov et al. [7] have fabricated new composites with enhanced thermal conductivity (500–900 W/(mK)) and mechanical properties. Ciupiński et al. [5] have prepared diamond/Cu composites including Cr_3_C_2_ carbide via pulse plasma sintering, which shows maximum thermal conductivity of 687 W/(mK). Using magnetron sputtering and a time-domain thermoreflectance (TDTR) technique, Chang et al. [2] have compared the thermal conductance of Cu/Ti/diamond and Cu/TiC/diamond structures, which proved that the formed carbide layer rather than the metallic layer is responsible for the impressive enhancement of the thermally conductive properties. For boron, it can be directly alloyed into Cu matrix or grown on diamond surface as coating. The formation of B_4_C was observed in the preparation processes of surface metallization and matrix alloying. Compared with the system without B addition, the tensile strength and thermal conductivity were all increased [3,4]. For the optimal boron content of 0.8 wt%, the highest thermal conductivity reaches 538 W/(mK) [8]. Except the traditional carbides, an in situ graphene interlayer has also been employed to enhance interface thermal conductance and reduce acoustic mismatch between diamond and copper [9]. The mechanisms of interface thermal resistance reduction by element additions have been investigated by researchers. Jia et al. [10] mainly studied the interface structure of diamond/Cu composite materials after adding Cr and its effect on heat conduction. The authors proposed that the difference between the degree of interface mismatch and the Debye temperature affects the transmission of phonons and causes the thermal resistance of the interface to increase. The authors believed that the addition of B_4_C coating gave rise to a dense composite. Elaborating the interface thermal conductance problem to increase the understanding of thermal transferring mechanisms will help to improve the thermal conductivity of the diamond/Cu system in the future [11].

Previous theoretical research has mainly concentrated on the acoustic mismatch model (AMM) [2,5] or diffuse mismatch model (DMM) [12]. However, the carriers of heat in diamond and copper are phonons and electrons, respectively. The actual interface structures are not taken into account by these two models. Scattering and phonon–electron interactions at the interface determine the interface thermal conductance of diamond/Cu composites. Thus, the atomic and electronic structures of the interfaces play a vital role in thermal conductance. First-principles methods within the framework of density–functional theory (DFT) are an effective and powerful tool to address the physical and chemical properties of the interfaces [13,14,15,16,17]. For examples, Guo et al. [13] have studied the probable fracture behaviors of diamond/Cu interfaces by the work of separation and the interface energy and indicated that fracture might occur in the metal phase near the interface. Recently, the influence of Mo doping on the mechanical properties and thermal conductivity of the interface was discussed [17]. Researchers have theoretically proved that Mo is a good additive element for the improvement of thermal conductivity by the analyses of electronic structures.

Although a great deal of work has been carried out on the interface regulation and properties of diamond/Cu composites, the related micro-mechanisms of interface formation and thermal conductance promotion still need to be further unraveled. In this paper, the interfacial carbides formation process and the enhancement mechanisms of interfacial heat conduction in diamond/Cu-B composites were studied by HRTEM and first-principles calculations. Boron occupation and diffusion behavior in Cu-B alloys and the segregation behavior of boron at diamond/Cu interfaces were revealed. This study provides valuable insights into the design of high thermal conductivity microzone structures in diamond/Cu composites.

## 2. Materials and Methods

### 2.1. Experiments

Diamond/Cu composites with 60 vol.% diamond content were prepared by pressure infiltration. Pressure infiltration was performed to prepare diamond/Cu composites with different boron content. Copper bulks and boron powder (99.9% in purity, size 2–3 μm and purity 99.99 wt%, Cuibolin Non-Ferrous Technology Co., Beijing, China) were used as the composite matrix. Single-crystal diamond with grain size of 100 μm produced by Huanghe Whirlwind Co. was used as the reinforcing phase. Firstly, diamond particles are made into preforms and placed in graphite molds. Using vacuum pressure infiltration, Cu-B alloy (0 wt.%B, 0.5 wt.%B and 1 wt.%B) is poured into the graphite mold at 1250 °C. Mechanical pressure is applied to force copper to penetrate into the preform gap. Then diamond/Cu composites were cooled to room temperature.

The microstructures of diamond/Cu interfaces were analyzed using scanning electron microscopy (SEM, JSM-7610F Plus, Hitachi, Tokyo, Japan) and a transmission electron microscope (TEM; JEOL 2100f, Tokyo, Japan). X-ray diffraction patterns were collected by X-ray diffraction (XRD; X’Pert-Pro MPD, Holland Panalytical, Almelo, The Netherlands) with Cu Kα. The thermal conductivity of diamond/Cu composites can be calculated by the formula *K = α·ρ_c_·C_p_*. The *α*, *ρ_c_* and *C_p_* respectively represent the thermal diffusivity, specific heat and density of the composite. The diamond/Cu composites’ thermal diffusivities were measured by a thermal conductivity tester (LFA447, NETZSCH, Selb, Germany) at room temperature.

### 2.2. DFT Calculations

The total energy and electronic structure calculations were carried out by employing the Vienna ab initio simulation package (VASP) [18,19]. The projector augmented wave (PAW) method [20,21] was used to describe the electron–ion interaction, and the electron exchange and correlation were treated within the generalized gradient approximation (GGA) in the Perdew-Burke-Ernzerhof (PBE) form [22]. The cut-off energy for the basis was set to be 500 eV for all the systems.

The structures of diamond-cubic lattice of diamond and *fcc* lattice of copper are depicted in Figure 1a,b. The relaxed lattice constants we obtained for these phases were 3.63 and 3.57 Å, respectively. To study the boron behavior in bulk copper, we built a 3 × 3 × 3 supercell consisting of 108 atoms, and a sufficiently accurate *k-*mesh (5 × 5 × 5) in a Monkhorst-Pack scheme for the integration over the Brillouin zone was employed.

For the copper(111)/diamond(111) interfaces, we built the heterointerfaces by connecting the seven copper layers and six carbon bilayers with a vacuum layer of 10 Å along the nonperiodic direction as depicted in Figure 1d. To deal with segregated atoms at low concentration, a (2 × 2) extended cell was adopted for the interface model. The corresponding *k-*mesh was 7 × 7 × 1. As copper was supposed to be constrained to the diamond lattice, we could obtain a relatively small mismatch, about 1.65%.

To depict boron atom diffusion behavior, the nudged elastic band (NEB) method was adopted with five images linearly interpolated between the initial and final points on the migration track. In all calculations, the supercell parameters were fixed and total energy convergence to less than 10^−4^ eV was set during the optimization of the atomic positions.

Besides, to address the impacts on thermal conductivity, the electron and phonon contributions on thermal conductance at interfaces have been discussed. For electrons, the transmission coefficients and electrical conductance of the diamond/Cu interfaces were calculated using the Nanodcal software package [23] in the framework of the nonequilibrium Green’s function–density functional theory (NEGF-DFT) [24]. For phonons, a variety of phonon density of states (PDOS) were obtained by the Cambridge Serial Total Energy Package (CASTEP) [25,26].

## 3. Results and Discussion

### 3.1. Microstructures and Phase Constitutions

The fractured surfaces of diamond/Cu and diamond/Cu-B composites are characterized by SEM in Figure 2a–c, where it can be observed that diamond particles are extracted from the matrix in the composite without boron addition. This phenomenon indicates that the interface bonding between diamond and matrix is too weak. With the addition of 0.5 wt.% boron, more transgranular fracture diamond particles could appear in the SEM images. Moreover, a large number of honeycomb substrates are attached to the surfaces of the extracted diamond particles in Figure 2c. When diamond/Cu composites break, the crack preferentially propagates along the defect. The stress concentration leads to diamond transgranular fracture. However, when the bonding between matrix and B_4_C is weak, diamond particles will be debonded. The microstructure analyses demonstrate that the weak interfacial bonding between diamond and copper matrix might be improved by boron addition. Further, in comparing the microstructures of different holding times, increasing the holding time could weaken the interface bonding.

For diamond/Cu-B composites, the phase compositions of the polished surfaces (Figure 2d) and the elemental composition of the near-interface region are measured by XRD and EDS. As shown in Figure 2e,f, significant boron accumulation can be detected near the diamond interfacial region. This indicates that the boron element in the copper–boron solid solution diffuses to the diamond during the infiltration process. Further, phase analyses have showed that a newly formed phase of B_4_C is found except for copper phase and diamond phase in Figure 2f. B_4_C can be confirmed to be the product of interfacial reaction during the diamond/Cu-B composites infiltration process. The presence of B_4_C enhances the interfacial bonding between diamond and copper inferred from the fracture morphology.

The microstructure at the interface between diamond and copper is shown in Figure 3. The surface of diamond particles stripped from the diamond–copper composite is covered with B_4_C. The structure of B_4_C extends from the diamond surface to the copper substrate in a tooth shape. TEM was used to observe the interface between diamond and B_4_C. It is found that the diamond surface has a nano-cone shape. The above-mentioned unique structure is beneficial to the transport efficiency of phonons. The formation of this special interface structure is revealed by first-principles calculation.

### 3.2. The Formation Kinetics and Thermodynamics of Carbide Interlayer B_4_C

The carbide interlayer B_4_C can be formed by both surface metallization of diamond and copper matrix alloying in diamond/Cu-B composites. However, the latter could produce more complex phase compositions and microstructures, i.e., copper, diamond, Cu-B solid solution, carbide interlayer B_4_C and the interfaces between these phases. From the atomic level, the complexity of the structures is mainly determined by the behavior of boron. It can diffuse and segregate to the interface region from Cu-B solid solution, consequently resulting in some boron atoms forming B_4_C interlayer. In order to discuss the formation of carbide B_4_C at the diamond/Cu-B interface, the boron occupation, diffusion, self-trapping and segregation behavior in diamond/Cu-B composites will be simulated at the atomic level.

Firstly, the boron preferred occupation site is investigated by DFT calculations. The stability for a single boron atom in Cu-B alloys can be determined by its formation energy:(1)$$E_{f}=E_{system}^{\mathrm{tot}}-n\mu_{\mathrm{Cu}}-m\mu_{B}$$
where $E_{system}^{\mathrm{tot}}$ is the total energy of the system with boron solute atoms, *n* and *m* are the numbers of copper and boron atoms in the system, respectively, and $\mu_{\mathrm{Cu}}$ and $\mu_{B}$ are the corresponding chemical potentials. Here, the bulk copper and boron are taken as references. We consider three possible positions of the single boron atom, namely, the substitutional site (SUB) and two interstitial sites, which including the tetrahedral interstitial site (TIS) and the octahedral interstitial site (OIS). The calculated formation energies for each case are summarized in Table 1. The results show that boron atoms could be energetically favorable to occupy the OISs with the formation energy of 1.63 eV. Boron atoms at the TISs are less stable than those at the substitutional sites.

In order to elucidate the kinetic behavior of boron in Cu-B composites, the diffusion and migration paths of boron in bulk copper are examined. Here, we only consider the barrier along the migration path through the relatively stable interstitial site by using the NEB method. According to the formation energies obtained above, the initial and final positions for the diffusion paths are both OISs in copper. The boron migration path 1→2→3 is also marked with the black arrows as shown in Figure 4. The results demonstrate that an interstitial boron atom has to overcome an energy barrier of 0.87 eV in order to diffuse between OISs.

To further understand the interactions between the boron atoms, we place two B in the nearest neighboring OISs with the initial distance of 2.57 Å. After relaxation, it was noticed that two B atoms could finally bind with each other due to the strongly negative binding energy (~−0.31 eV) with the final distance of 1.74 Å. This characteristic distance is almost equivalent to the lengths of the B-B bonds (1.76~1.83 Å) in the compound B_4_C. To set the two boron atoms farther apart, this strong attraction quickly disappeared. Similarly, it is found that the binding energy of B-C atoms in the nearest neighboring OISs is about 0.18 eV with the final distance of 2.83 Å, larger than the B-C bonds (1.66 Å) in the compound B_4_C. Thus, the B-B self-trapping interactions mainly cause the atom aggregation in materials.

The underlying mechanisms of the interactions in the Cu-B systems can be illuminated by the analyses of the electronic structures. The calculated density of states (DOS) is plotted for pure copper, one boron atom at OIS (Cu_108_B) and two boron atoms at the nearest OISs (Cu_108_B_2_) in Figure 5a. An isolated boron atom is mainly composed of localized *s* and *p* states, and its *s* states are lower than its *p* states. We can see when a boron atom occupies the OIS that DOS for Cu-d are slightly changed and the B-p states are largely broadened for both conduction and valence bands. Comparatively, the B-s states still keep localized at the deep level. This demonstrates that the solo boron atom at the OIS suffers from compression, which affects the B-p states more severely. In Figure 5b,c, we compare the calculated density of valence electrons between Cu_108_B and Cu_108_B_2_ systems at the Cu-B (100) planes. The most noteworthy feature is that there is a charge bridge between two boron atoms, indicating an existing strong covalent bond between boron atoms. In addition, when two boron atoms at the nearest OISs are put together, the B-s states shift toward a deeper level, leading to a more stable system.

According to our experimental results, the segregation behavior of boron at diamond/Cu interfaces is very critical for the formation of carbide interlayer B_4_C in diamond/Cu-B composites. To study the segregation behavior, one boron atom will be placed in the bulk (Site A) and near the interface (Sites B and C) as shown in Figure 1e. The segregation energy *E*^seg^ can be calculated as follows:$$E^{\mathrm{seg}}=E^{\mathrm{tot}}\left(\mathrm{interface},B/C\right)-E^{\mathrm{ref}}\left(\mathrm{bulk},\;A\right),$$
where the positive or negative value means a boron atom favors staying away from or segregating to the interfaces. For Sites B and C, the segregation energies we calculated are −0.13 and −1.10 eV, respectively. This means that the boron atom is very energetically favorable to segregate to diamond/Cu interfaces. With a powerful driving force (~−1.10 eV), it is likely to pass through the copper layers and be bonded with carbon atoms at the interfaces. These simulation results are consistent with our experiments.

The underlying mechanisms of the segregation behavior can be explained by the analyses of the DOS as shown in Figure 6. For pure diamond/Cu interfaces, Cu-*d* states hybridizes with C-*p* ones. An peak appear at the Fermi level implies that the interface bonding of the composite is weak. As for boron-segregated diamond/Cu interfaces, the B-*p* states become more delocalized because of the strong hybridization with C-*p* states. Also, the decrease of the peak value at the Fermi level demonstrates that the interface interaction is enhanced.

In addition, the formation of a carbide interlayer B_4_C in experiments can be understood by thermodynamics. B_4_C has the rhombohedral structure with a D_3d_^5^-R3m space group, as shown in Figure 1c. The optimized lattice parameters are a = 5.19 and c = 12.12 Å. Then, the formation enthalpy of B_4_C under such a growth condition can be calculated by:$$E_{f}=E^{\mathrm{tot}}\left(B_{4}C\right)-4\mu_{B}-\mu_{C}.$$

In the above equation, $E^{\mathrm{tot}}\left(B_{4}C\right)$ represents the total energy of B_4_C unit cell, $\mu_{B}$ and $\mu_{C}$ are the chemical potentials of boron and carbon. Here, the reference states are per a carbon atom in diamond and a boron atom at OIS of the bulk copper. Therefore, the calculated formation enthalpy is −1.36 eV/atom, which is even lower than the segregation energy of boron at interfaces. Thus, we can safely say that the reaction of B+C→B_4_C at interfaces is exothermic.

### 3.3. The Enhancement Mechanisms of Interfacial Heat Conduction

As listed in Table 2, the thermal conductivity for the diamond/Cu, diamond/Cu-0.5 wt.%B and diamond/Cu-1.0 wt.%B are 261 W/(mK), 694 W/(mK) and 647 W/(mK), respectively. After adding boron, the density and specific heat of diamond/Cu composites remain basically unchanged, but the thermal diffusion coefficient is greatly improved. The thermal conductivity of pure copper and diamond are 390 W/mK and 1500 W/mK, respectively. The results show that reducing the interface thermal resistance by adding boron can improve the thermal conductivity of diamond/Cu composites by 2.5 times. Because the thermal conductivity of boron carbide is much lower than that of copper and diamond, the content of boron carbide at the interface has a significant effect on the composites’ thermal conductivity. When the boron content increases to 1.0 wt.%, the thickness of B_4_C at the interface increases. The thermal conductivity of composites decreases due to the increase of interfacial thermal resistance. Therefore, adding excessive boron will lead to the decrease of thermal conductivity.

It is well known that the carriers of heat in diamond and copper are phonons and electrons, respectively. Scattering and phonon–electron interactions at the interface play a pivotal role in determining the interface thermal conductance. As depicted in Figure 7a, the electron states at the Fermi level might be beneficial to the heat transfer. However, the previous research by Liu et al. [21] demonstrated that the metallic state is mainly characterized on the first diamond layer at diamond/Cu interfaces, and thus the electron contribution on the heat transfer is relatively small. In order to quantitatively address the electron contribution, we have calculated electronic transport properties in the systems. The transmission coefficients shown in Figure 7b,c represent the energy scattering states traveling from left lead to right lead through the scattering region. The eigenstates of pure Cu possessing higher transmission overlap in *k*-space (See Figure 7b) indicate the higher electron transport efficiency as compared with that of the Cu-diamond-Cu contact (see Figure 7c), which is consistent with the comparison of calculated conductance. The conductance of diamond/Cu interfaces is dramatically lower (g_interface_ = 1.13 G_0_/nm^2^) than the conductance density of Cu metal (g_Cu_ = 13.63 G_0_/nm^2^). These results verify that the electron contribution on heat transfer is relatively insignificant.

According to DMM theory, the ratio of phonon conduction from one pattern of material to another depends on the matching degree of phonon spectra of the two materials that make up the interface. Usually, the phonon DOS is utilized to illustrate the thermal transport behavior across an interface, and the higher overlap of the phonon DOSs means more effective phonon coupling at the interface [27,28]. To illustrate the phonon coupling at boron-segregated diamond/Cu interfaces, the phonon DOS is calculated for bulk diamond and Cu, B_4_C compound and Cu_4_B solid solution (see Figure 8a). It should be noted that phonon states of bulk Cu are located in the frequency range of 0~8 THz, which are lower than those (10~40 THz) of bulk diamond. The huge difference causes the phonons to encounter higher resistance when transferring heat at the interfaces. The predicted poor phonon coupling is consistent with the low experimental thermal conductivity of the diamond/Cu composites. The formation of B_4_C at the interface can establish a phonon transmission bridge between diamond and copper owing to its phonon frequencies from 5 to 34 THz. As shown in Figure 8b, both boron and carbon atoms equally contribute to the phonon spectrum from low to high frequencies. Its distribution both matches diamond well in high frequencies and overlaps with Cu in low frequencies.

On the other hand, diamond/Cu-B composites were prepared by matrix alloying in experiments. It is noteworthy that some boron atoms diffuse to the interface area and react with carbon to form B_4_C at high temperatures, and the other still exist in tetrahedral interstitial sites of copper. So the effect of Cu-B solid solution on the thermal conductivity should not be neglected. Here, we take Cu_4_B solid solution as an example. As depicted in Figure 8, the formation of Cu_4_B solid solution splits the main peak of Cu and makes it broadened to higher frequencies because of the volumetric expansion effect. Also, we can observe a new peak at the middle frequencies (17~20 THz), which is majorly produced by boron atoms. For Cu, the carriers of heat are mainly electrons. Thus, it might be expected that the superior thermophysical properties in the material are mainly attributed to the formation of B_4_C. In all, the addition of boron can improve the interfacial heat transfer, thereby increasing the thermal conductivity of diamond/Cu composites.

## 4. Conclusions

To obtain high interface thermal conductivity and reveal an interface microscopic heat transfer mechanism, the interfacial carbides formation process and the enhancement mechanisms of interfacial heat conduction in diamond/Cu-B composites were studied by HRTEM and first-principles calculations. The main conclusions are summarized as follows:Diamond/Cu-B composites achieved the highest thermal conductivity of 694 W/mK with a boron addition of 0.5 wt.%. The addition of boron improves the interface bonding, thus increasing the thermal conductivity of diamond/Cu composites.The calculation results demonstrate that boron can diffuse toward the interface region with an energy barrier of 0.87 eV, and these boron elements are energetically favorable to form the B_4_C phase. By adjusting the preparation process and boron concentration, the interface between B_4_C and diamond can be transformed from a straight interface to a dentate interface.The B_4_C phonon spectrum is distributed in the range of the copper and diamond phonon spectrum. The overlapping of phonon spectra and the dentate structure together enhance the interface phononic transport efficiency.

## Figures and Tables

**Figure 1 materials-16-02021-f001:**
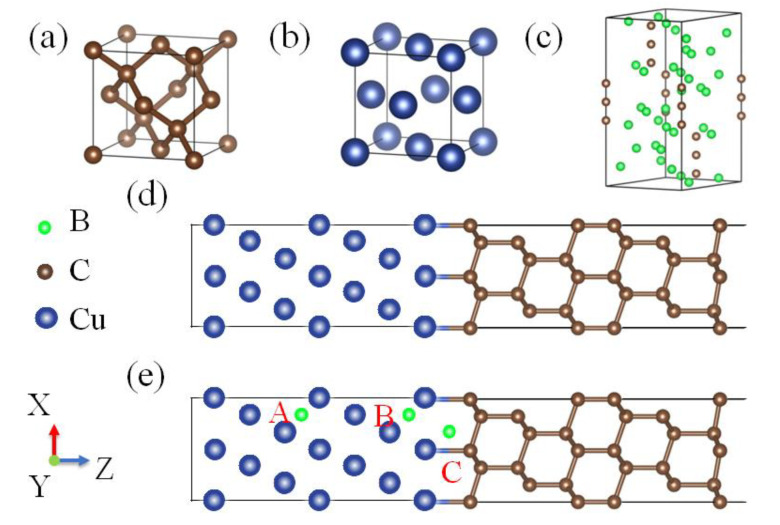
The structures of: (**a**) diamond-cubic lattice of diamond; (**b**) *fcc* lattice of copper; (**c**) B_4_C; and (**d**,**e**) diamond/Cu interfaces without and with boron segregation. A, B and C denote different sites for boron.

**Figure 2 materials-16-02021-f002:**
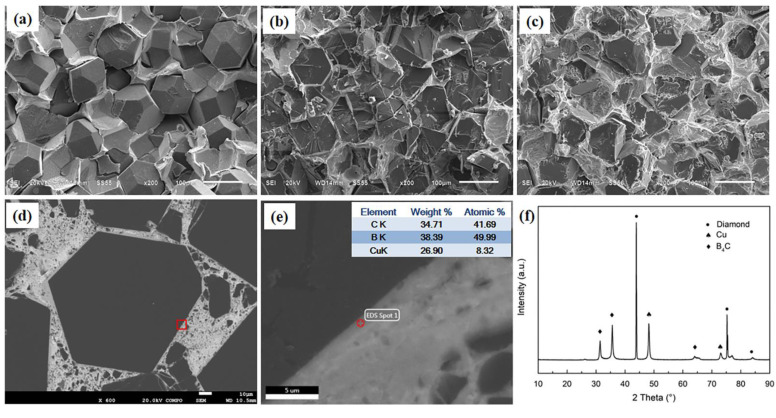
SEM images for the fractured surfaces of: (**a**) diamond/Cu; (**b**) diamond/Cu-B composites with 0.5 wt.% boron; (**c**) diamond/Cu-B composites with 1.0 wt.% boron; (**d**) polished surface of diamond/Cu-B composites (**b**); (**e**) element analysis of interface area in (**d**); and (**f**) XRD patterns of diamond/Cu-B composites (**b**).

**Figure 3 materials-16-02021-f003:**
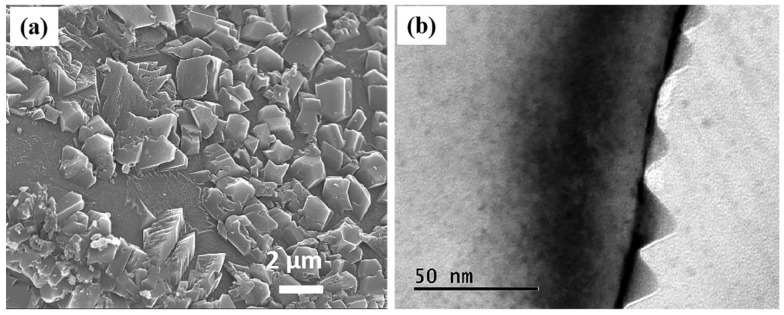
(**a**) B_4_C microstructure at Diamond-Cu interface; and (**b**) microstructure at diamond-B_4_C interface.

**Figure 4 materials-16-02021-f004:**
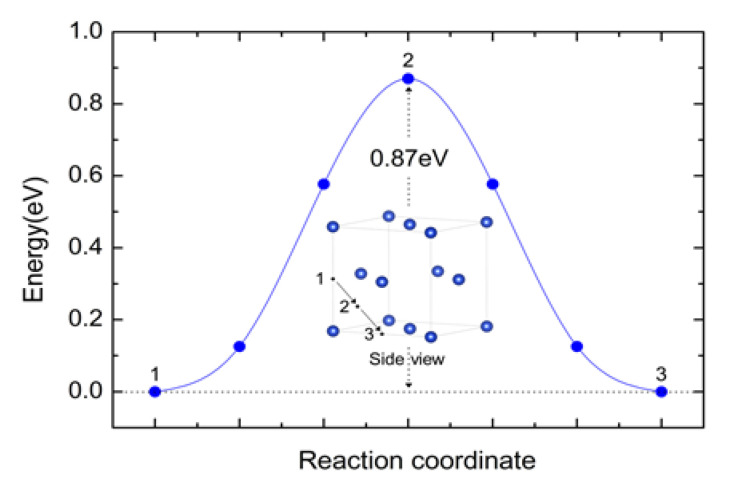
Diffusion energy profile and the diffusion paths between OISs for boron in copper. Site 1 and 3 denote the nearest OISs. Site 2 is the transitional state.

**Figure 5 materials-16-02021-f005:**
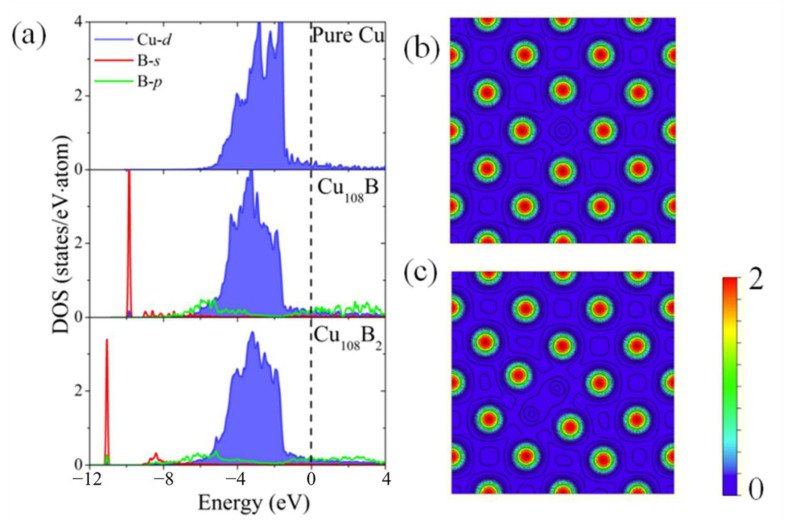
(**a**) Density of states (DOS) for pure copper, one boron atom at OIS (Cu_108_B) and two boron atoms at the nearest OISs (Cu_108_B_2_). The dotted line represents the Fermi level. (**b**,**c**) The valence charge densities of the (001) plane for Cu_108_B and Cu_108_B_2_, respectively. Contours start from 0.025 e/a.u.^3^, and increase successively by a factor of 10^1/5^.

**Figure 6 materials-16-02021-f006:**
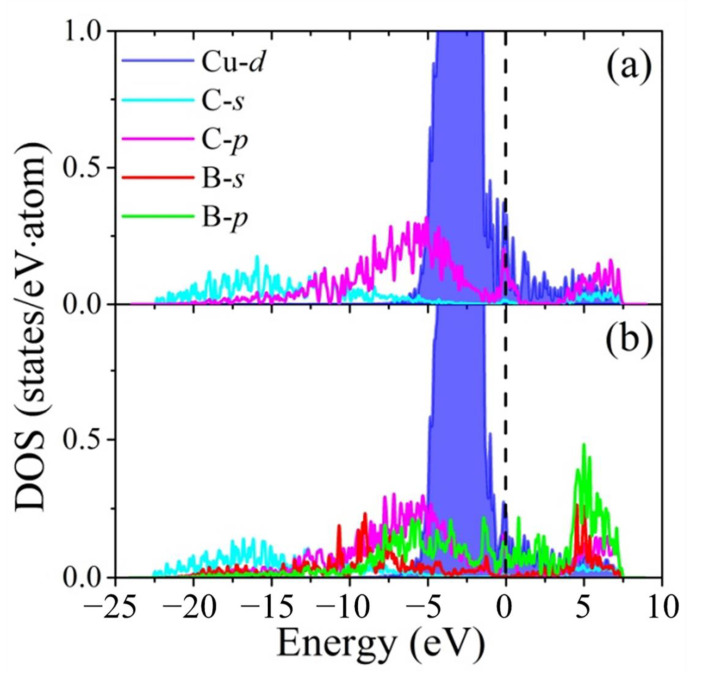
Density of states (DOS) for (**a**) pure and (**b**) boron-segregated diamond/Cu-B interfaces. The dotted line represents the Fermi level.

**Figure 7 materials-16-02021-f007:**
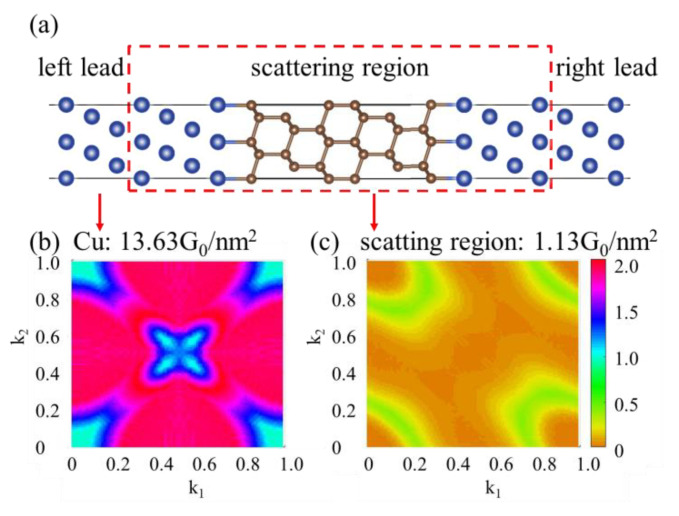
(**a**) Side view of the atomic structure of the diamond/Cu interfaces in NEGF-based electronic transport calculations. The dashed red line delimits the simulation box for the ab initio transport calculations; periodic boundary conditions are applied in the plane perpendicular to transport. (**b**,**c**) present the transmission coefficients (averaged on energy) in reciprocal space of pure Cu and diamond/Cu interfaces respectively, as well as the total conductance per unit area.

**Figure 8 materials-16-02021-f008:**
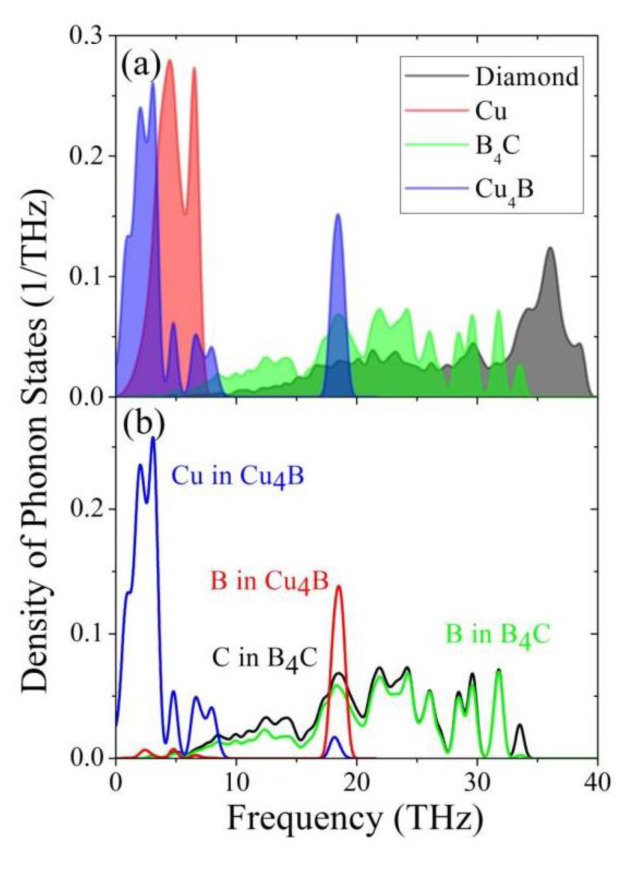
(**a**) Phonon density of states (PDOS) for bulk diamond and Cu, B_4_C compound and Cu_4_B solid solution. (**b**) The contribution for per atom is also included.

**Table 1 materials-16-02021-t001:** The calculated formation energies (eV) of an isolated boron atom doped in copper.

	B_SUB_	B_OIS_	B_TIS_
This work	1.90	1.63	3.09

**Table 2 materials-16-02021-t002:** Thermal physical properties of diamond/Cu and diamond/Cu-B composites.

	Density (g/cm^3^)	Specific Heat Capacity (J/(gK))	Thermal Diffusivity Coefficient (mm^2^/s)	Thermal Conductivity (W/(mK))
Diamond/Cu	5.52	0.46	102.95	261
Diamond/Cu-0.5 wt.%B	5.73	0.46	263.30	694
Diamond/Cu-1.0 wt.%B	5.75	0.46	244.76	647

## Data Availability

Not applicable.

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
