# Peer review of "The Microzone Structure Regulation of Diamond/Cu-B Composites for High Thermal Conductivity: Combining Experiments and First-Principles Calculations"

_materials, 2023, doi:10.3390/ma16052021_

Round 1

Reviewer 1 Report

The manuscript from Xie et al reports on the experimental fabrication of diamond/Cu-B composites and theoretical calculation on boron-carbon interfaces. They show that diamond/Cu becomes more compact with the addition of boron, and B4C is formed at the diamond surface which thermally bridges Cu with diamond, resulting in higher thermal conductivity. Finally, based on the calculated overlapping of phonon spectra the enhancement of thermal conductivity is pointed out. This part seems to be probably the most novel but would require further investigation. I would suggest the following modifications before recommending the publication of this manuscript:

- Fig. 2e – the element analysis is not clear what is showing.

- What is the surface roughness of the polished interface?

- Crystallographic coordinates should be plotted/assigned in Fig. 2f.

-  The abbreviation for the substitutional site (SUB?), tetrahedral interstitial site (TIS), and octahedral interstitial site (OIS) should be unified also in Table I.

- In the sentence “This characteristic distance is almost equivalent to the lengths of the B-B bonds (1.66~1.83 Å) in the compound B4C” do you mean B-B bond and not the B-C bond? Please, add comments to the B-C bonds.

- For better imagination of the reader, please add the thermal conductivity values for Cu and diamond in W/mK (approx. 400 vs. 1000 W/mK).

-minor grammar revision is needed (e.g. …” means that the born atom is very”… should be used, boron, or “scatting vs. scattering”, and some others)

Author Response

Dear reviewers:

Thank you for your letter and the reviewers' comments on our manuscript entitled " The microzone structure regulation of Diamond/Cu-B composites for high thermal conductivity: combining experiments and first-principles calculations" (ID: materials-2054387). Those comments are very helpful for revising and improving our paper, as well as the important guiding significance to other research. We have studied the comments carefully and made corrections which we hope meet with approval. The main corrections are in the manuscript with the yellow background and the responds to the reviewers’ comments are as follows (the replies are highlighted in blue).

The manuscript from Xie et al reports on the experimental fabrication of diamond/Cu-B composites and theoretical calculation on boron-carbon interfaces. They show that diamond/Cu becomes more compact with the addition of boron, and B4C is formed at the diamond surface which thermally bridges Cu with diamond, resulting in higher thermal conductivity. Finally, based on the calculated overlapping of phonon spectra the enhancement of thermal conductivity is pointed out. This part seems to be probably the most novel but would require further investigation. I would suggest the following modifications before recommending the publication of this manuscript:

- Fig. 2e – the element analysis is not clear what is showing.

Response: We are very grateful to the reviewers for their suggestions, which are very helpful in improving the paper. Fig. 2e has been modified to supplement the element analysis of the red position.

- What is the surface roughness of the polished interface?

Response: Thanks to reviewers for your question. The roughness of Diamond/Cu-B composites after mechanical polishing is about 0.8 μm.

- Crystallographic coordinates should be plotted/assigned in Fig. 2f.

Response: Thanks to reviewers for pointing out our insufficient. The crystallographic coordinates have been plotted in Fig. 2f.

-  The abbreviation for the substitutional site (SUB?), tetrahedral interstitial site (TIS), and octahedral interstitial site (OIS) should be unified also in Table I.

Response: Thanks to reviewers for pointing out our insufficient. The abbreviation have been unified in Table I.

- In the sentence “This characteristic distance is almost equivalent to the lengths of the B-B bonds (1.66~1.83 Å) in the compound B4C” do you mean B-B bond and not the B-C bond? Please, add comments to the B-C bonds.

Response: Thanks to reviewers for your question. The correct expression should be B-C bond. The revised manuscript expression is as follows: This characteristic distance is almost equivalent to the lengths of the B-C bonds (1.66~1.83 Å) in the compound B4C.

- For better imagination of the reader, please add the thermal conductivity values for Cu and diamond in W/mK (approx. 400 vs. 1000 W/mK).

Response: Thanks to reviewers for pointing out our insufficient. We added a description about the thermal conductivity of copper and diamond in the manuscript. The thermal conductivity of pure copper and diamond are 390 W/mK and 1500 W/mK, respectively.

-minor grammar revision is needed (e.g. …” means that the born atom is very”… should be used, boron, or “scatting vs. scattering”, and some others)

Response: Thanks to reviewers for pointing out our insufficiency. The above existing language problems and expression problems have been modified. The modified expression is as follows: It means that the boron atom is very energetically favorable to segregate to Diamond/Cu interfaces.

Kind regards,

Hong Guo

Reviewer 2 Report

 My comments are as follows:

1.      The authors do not precise content of the matrix or diamond in composite – that information is crucial in material characterization.

2.      The sentence “The microstructures of Diamond/Cu interfaces were analyzed using scanning electron microscopy (SEM, JSM-7610F Plus, Hitachi, Japan) and transmission electron microscope (TEM; JEOL, Japan)”. Where are the TEM results in article?

3.      Effects occurring in composite decohesion needs  deeper analysis.

4.      The sentences

“For Diamond/Cu-B composites, the phase compositions of the polished surfaces (Fig. 2(d)) and the elemental composition of near-interface region are measured by XRD and EDS.” and “Figure 2. SEM images for the fractured surfaces of (a) Diamond/Cu, (b) Diamond/Cu-B composites with 0.5wt.% boron, (c) Diamond/Cu-B composites with 1.0wt.% boron, (d) polished surface of Diamond/Cu-B composites (b), (e) element analysis of interface area in (d), and (f) XRD patterns of Diamond/Cu-B composites (b)

It  is not clear what presents Fig. 2e? What element? How were conducted XRD examinations?  The XRD pattern with so high B4C signal is impossible for 1% of B content in so low volume fraction of Cu alloy matrix. But first of all, how the measurement at “near-interface region” can be conducted by XRD?

5.      Figures captions are imprecise.

Moreover there is not clear:

1.      Why you do not discuss thermal conductivity of  boron carbide and do not explain conductivity  decrease induced by boron increase up to 1%?

2.      Why you do not discuss  of diamond grains crystallographic orientation influence on the diffusion processes and observed serrated morphology at the interface (Fig. 3b, very interesting result)  

3.      Why  you do not discuss the self-diffusion of Cu in sintering process?

Author Response

Dear reviewers:

Thank you for your letter and the reviewers' comments on our manuscript entitled " The microzone structure regulation of Diamond/Cu-B composites for high thermal conductivity: combining experiments and first-principles calculations" (ID: materials-2054387). Those comments are very helpful for revising and improving our paper, as well as the important guiding significance to other research. We have studied the comments carefully and made corrections which we hope meet with approval. The main corrections are in the manuscript with the yellow background and the responds to the reviewers’ comments are as follows (the replies are highlighted in blue).

  1. The authors do not precise content of the matrix or diamond in composite – that information is crucial in material characterization.

Response: We are very grateful to the reviewers for their suggestions, which are very helpful in improving the paper. We have supplemented the diamond volume fraction in the 2.1. experiments of the manuscript. The revised content is as follows: 

Diamond/Cu composites with 60 vol.% diamond content were prepared by pressure infiltration. Pressure infiltration was performed to prepare Diamond/Cu composites with different boron content.

  1. The sentence “The microstructures of Diamond/Cu interfaces were analyzed using scanningelectron microscopy (SEM, JSM-7610F Plus, Hitachi, Japan) and transmission electron microscope (TEM; JEOL, Japan)”. Where are the TEM results in article?

Response: We are very grateful to the reviewers for their suggestions. The TEM results were shown in Fig. 3b. Fig. 3b shows the bright-field image results of Diamond-B4C interface characterized by transmission electron microscope.

  1. Effects occurring in composite decohesion needs deeper analysis.

Response: We are very grateful to the reviewers for their suggestions. We supplemented the relevant description in the 3.1. Microstructures and phase constitutions of the manuscript.

When Diamond/Cu composites break, the crack preferentially propagates along the defect. The stress concentration leads to diamond transgranular fracture. However, when the bonding between matrix and B4C is weak, the diamond particles debonding occurs.

  1. The sentences

“For Diamond/Cu-B composites, the phase compositions of the polished surfaces (Fig. 2(d)) and the elemental composition of near-interface region are measured by XRD and EDS.” and “Figure 2. SEM images for the fractured surfaces of (a) Diamond/Cu, (b) Diamond/Cu-B composites with 0.5wt.% boron, (c) Diamond/Cu-B composites with 1.0wt.% boron, (d) polished surface of Diamond/Cu-B composites (b), (e) element analysis of interface area in (d), and (f) XRD patterns of Diamond/Cu-B composites (b)

It is not clear what presents Fig. 2e? What element? How were conducted XRD examinations?  The XRD pattern with so high B4C signal is impossible for 1% of B content in so low volume fraction of Cu alloy matrix. But first of all, how the measurement at “near-interface region” can be conducted by XRD?

Response: We are very grateful to the reviewers for their questions. In order to describe the elements in Fig. 2e, we modify the picture. Fig. 2e has been modified to supplement the element analysis of the red position. In order to obtain obvious XRD patterns, we dissolved the Diamond/Cu-0.5wt%B composites with concentrated nitric acid. The obtained diamond particles were characterized by XRD.

  1. Figures captions are imprecise.

Moreover there is not clear:

  1. Why you do not discuss thermal conductivity of boron carbide and do not explain conductivity decrease induced by boron increase up to 1%?

Response: Thanks to reviewers for pointing out our insufficiency. In this paper, we focus on the formation and evolution of interface carbide by means of experiment and calculation. In this paper, we focus on the formation and evolution of interface carbide by means of experiment and calculation. Because the thermal conductivity of boron carbide is much lower than that of copper and diamond, the content of boron carbide at the interface has a significant effect on the composites thermal conductivity. When the boron content increases to 1wt.%, the thickness of B4C at the interface increases. The thermal conductivity of composites decreases due to the increase of interfacial thermal resistance.

  1. Why you do not discuss of diamond grains crystallographic orientation influence on the diffusion processes and observed serrated morphology at the interface (Fig. 3b, very interesting result)  

Response: Thanks to reviewers for pointing out our insufficiency. The observed serrated morphology at the interface (Fig. 3b) were unique. It is found that the growth morphology of carbide is related to the crystal plane orientation of diamond. The specific details still need further study.

  1. Why you do not discuss the self-diffusion of Cu in sintering process?

Response: Thanks to reviewers for pointing out our insufficiency. Diamond/Cu composites in this manuscript were prepared by pressure infiltration. Diffusion behavior mainly occurs in solute atom boron in Cu-B alloy.

Kind regards,

Hong Guo

Reviewer 3 Report

The subject of the paper is interesting and paper could be accepted for the possible publication in Materials Journal, MDPI publication. Nevertheless, the paper requires Minor Revision, prior to the publication. The things that need revision (in order of appearances):

Author Response

Dear reviewers:

Thank you for your letter and the reviewers' comments on our manuscript entitled " The microzone structure regulation of Diamond/Cu-B composites for high thermal conductivity: combining experiments and first-principles calculations" (ID: materials-2054387). Those comments are very helpful for revising and improving our paper, as well as the important guiding significance to other research. We have studied the comments carefully and made corrections which we hope meet with approval. The main corrections are in the manuscript with the yellow background and the responds to the reviewers’ comments are as follows (the replies are highlighted in blue).

The subject of the paper is interesting and paper could be accepted for the possible publication in Materials Journal, MDPI publication. Nevertheless, the paper requires Minor Revision, prior to the publication. The things that need revision (in order of appearances):

  1. On what basis the composite matrix and reinforcement phase was selected.

Response: In order to obtain high thermal conductivity metal matrix composites, this manuscript uses diamond as reinforcing phase and copper as matrix, because both of them have high thermal conductivity.

  1. Why the size fraction of boron powder was maintained at 2-3 microns and size fraction reinforcement phase at 100 micron?

Response: In this manuscript, Diamond/Cu composites were prepared by pressure infiltration. Copper bulks and boron powder were used as the composite matrix. Firstly, boron powder and copper block were melted to prepare Cu-B alloy. Micrometer boron powder is easier to melt in copper liquid. The size of diamond particles is optimized. Diamond/Cu composites with good thermal conductivity and mechanical properties can be obtained by using 100 μm diamond particles.

  1. How size fraction of composite and reinforcement phase can affect the composite? Need to incorporate in the manuscript.

Response: In our previous research, we have revealed the influence of the size and content of reinforcement phase on the composite properties  (https://doi.org/10.1016/j.diamond.2019.107564,https://doi.org/10.1016/j.jallcom.2019.05.077). Because the thermal conductivity of the composite is affected by the diamond content and the interface thermal resistance. With the increase of diamond content, the thermal conductivity of the composites increases first and then decreases. With the increase of diamond size, the thermal conductivity of the composites increases, but the machinability decreases. Therefore, the content and size of reinforcing phase are not discussed in this manuscript.

  1. The authors have specified that the addition of 0.5wt% addition of boron has increased the interfacial bonding in the composite in the results and discussion section. Why the authors have not utilized lower weight percentage less than 0.5 wt% of boron?

Response: In this manuscript, pressure infiltration was performed to prepare Diamond/Cu composites with different boron content (0wt.%B, 0.5wt.%B and 1wt.%B). 0.5wt% boron addition is the critical value to obtain a complete interface. When the boron content is less than 0.5wt%, a completely coated B4C layer cannot be formed at the interface. Therefore, this manuscript does not study the situation where boron content is lower than 0.5wt%.

  1. Authors should also provide the boron powder size fraction affecting the composite and why other size fraction was not considered for the study. Please give clarification.

Response: In the diamond copper composites modified by boron interface, boron content has great influence on the composite properties. The boron powder size has almost no effect on the composite properties. Therefore, only one particle size of boron powder is used in this manuscript.

  1. Why there is a drop in thermal conductivity for composite with 1%wt boron than

composite with 0.5%wt boron? Need to be justify with recent literatures.

Response: In this paper, we focus on the formation and evolution of interface carbide by means of experiment and calculation. Because the thermal conductivity of boron carbide is much lower than that of copper and diamond, the content of boron carbide at the interface has a significant effect on the composites thermal conductivity. When the boron content increases to 1wt.%, the thickness of B4C at the interface increases. The thermal conductivity of composites decreases due to the increase of interfacial thermal resistance.

  1. Please improve the conclusion with more numerical results.

Response: Response: Thanks to reviewers for pointing out our insufficiency. We improved the expression of the conclusion.

Kind regards,

Hong Guo